# Fluorocarbon Plasma-Polymerized Layer Increases the Release Time of Silver Ions and the Antibacterial Activity of Silver-Based Coatings

**DOI:** 10.3390/nano14070609

**Published:** 2024-03-29

**Authors:** Linda Bonilla-Gameros, Pascale Chevallier, Xavier Delvaux, L. Astrid Yáñez-Hernández, Laurent Houssiau, Xavier Minne, Vanessa P. Houde, Andranik Sarkissian, Diego Mantovani

**Affiliations:** 1Laboratory for Biomaterials and Bioengineering, (CRC-Tier I), Department of Min-Met-Materials Eng and Regenerative Medicine, CHU de Quebec, Laval University, Quebec City, QC G1V 0A6, Canadalidi.yanez-hernandez.1@ulaval.ca (L.A.Y.-H.); 2Laboratoire Interdisciplinaire de Spectroscopie Electronique, Namur Institute of Structured Matter, University of Namur, 61 Rue de Bruxelles, 5000 Namur, Belgium; xavier.delvaux@unamur.be (X.D.); laurent.houssiau@unamur.be (L.H.); 3Oral Ecology Research Group (GREB), Faculty of Dentistry, Université Laval, 2420 rue de la Terrasse, Quebec City, QC G1V 0A6, Canada; 4Plasmionique Inc., 171-1650 Boul Lionel Boulet, Varennes, QC J3X1S2, Canada; sarkissian@plasmionique.com

**Keywords:** silver nanoparticles, amorphous hydrogenated carbon, fluoropolymer, low pressure plasma

## Abstract

Silver-based antibacterial coatings limit the spread of hospital-acquired infections. Indeed, the use of silver and silver oxide nanoparticles (Ag and AgO NPs) incorporated in amorphous hydrogenated carbon (a-C:H) as a matrix demonstrates a promising approach to reduce microbial contamination on environmental surfaces. However, its success as an antibacterial coating hinges on the control of Ag^+^ release. In this sense, if a continuous release is required, an additional barrier is needed to extend the release time of Ag^+^. Thus, this research investigated the use of a plasma fluoropolymer (CF_x_) as an additional top layer to elongate Ag^+^ release and increase the antibacterial activity due to its high hydrophobic nature. Herein, a porous CF_x_ film was deposited on a-C:H containing Ag and AgO NPs using pulsed afterglow low pressure plasma polymerization. The chemical composition, surface wettability and morphology, release profile, and antibacterial activity were analyzed. Overall, the combination of a-C:H:Ag (12.1 at. % of Ag) and CF_x_ film (120.0°, F/C = 0.8) successfully inactivated 88% of *E. coli* and delayed biofilm formation after 12 h. Thus, using a hybrid approach composed of Ag NPs and a hydrophobic polymeric layer, it was possible to increase the overall antibacterial activity of the coating.

## 1. Introduction

Silver (Ag)-based antibacterial coatings have been developed in the last decade, becoming one of the most widely studied metal-based coatings owing to their potential in preventing and limiting the spread of hospital-acquired infections (HAIs, i.e., infections that patients acquire while receiving health care) [1,2]. Specifically, the use of Ag NPs embedded in amorphous hydrogenated carbon (a-C:H) coatings have been studied for their potential to delay or to inhibit bacterial colonization on environmental surfaces (i.e., non-intrusive soft or hard surfaces located in hospitals). Indeed, Ag NPs are known to be biologically active when they produce Ag ions (Ag^+^), exhibiting antibacterial activity against Gram-positive and Gram-negative bacteria [3,4]. The release of Ag^+^ has proven to be dependent mainly on the concentration and the size of Ag NPs, as well as their oxidation state [5]. However, there are still certain challenges that have delayed their impact on clinical practice. For instance, designing coatings that maintain released Ag^+^ levels within a concentration high enough to kill bacteria but low enough to limit cytotoxicity towards humans and the environment remains a significant task to overcome [6]. Not to mention that silver resistance determinants (i.e., resistance genes and mutations that give a microbe the ability to resist the effects of one or more drugs) are widely found among environmental and clinically relevant bacteria [7,8]. This is of concern because the extensive use of Ag-based products will increase the release of silver in the environment, potentially inducing the dissemination of silver resistance and, therefore, cross-resistance to antibiotics. Ultimately, their success depends on their ability to deliver precise doses within a proper timeframe (i.e., controlled release).

The deposition of an additional permeable polymeric top layer can act as a barrier to extend the duration of sustained-released Ag^+^. Particularly, fluoropolymers (CF_x_) in the form of thin films have attracted recent attention due to their outstanding tribological, wetting, and bioactive properties [9,10,11,12,13]. In fact, their inherent non-stick properties due to their low surface energy have shown to prevent the initial steps of bacterial adhesion on a substrate surface and, therefore, the subsequent formation of biofilm [14,15,16,17,18]. In addition, several studies have reported their application as a fluoropolymer matrix containing Ag NPs [19,20,21,22,23]. A promising approach to deposit these fluoropolymers is using low-pressure plasma polymerization, known to produce highly adherent coatings [24]. Due to the versatility of plasma polymers, other fluoropolymer designs have been proposed, like sandwich and multi-layered coatings [25,26]. Moreover, the thickness, the degree of crosslinking, inversely proportional to porosity, and the hydrophobicity of the fluoropolymer can be tuned; thus, the release profile can be controlled.

Therefore, the aim of this work was to investigate the influence of a plasma fluoropolymer as a top layer on amorphous hydrogenated carbon coatings containing Ag and AgO NPs on the release profile of Ag^+^. To the authors’ best knowledge, this is the first study where the combination of a porous and hydrophobic CF_x_ film is desired and used to modulate the release of Ag^+^ from Ag-based a-C:H coatings. Indeed, the combination of the above-mentioned components into a multi-layered coating emerges as a promising multi-approach strategy to increase their antibacterial activity without further increasing Ag concentration. To this end, Ag and AgO NPs were first simultaneously deposited on an a-C:H film using a hybrid low pressure plasma method, combining physical vapor deposition and chemical vapor deposition. The fluoropolymer was then deposited by pulsed afterglow plasma polymerization on a glow discharge low pressure plasma reactor. Special attention was focused on the porosity of the CF_x_ layer as the main factor to control the release of Ag^+^ [27]. Herein, a comprehensive report is presented on the chemical composition, the surface wettability and morphology, as well as the final release profile and antibacterial behavior of the coatings.

## 2. Experimental Section

### 2.1. Materials

The 100-oriented single silicon substrates were supplied by University Wafers (Boston, MA, USA). The substrates were cut into 1 cm^2^ and sequentially cleaned in an ultra-sonic bath with acetone (Thermo Fisher Scientific, Saint-Laurent, QC, Canada), deionized millipore water (DI, resistivity: 18 MΩ-cm), and methanol (Thermo Fisher Scientific, Saint-Laurent, QC, Canada) before plasma deposition. 

BD Bacto^TM^ Tryptic Soy Broth (TSB), BD Bacto^TM^ Dehydrated Agar, glycerol and glutaraldehyde were provided by Thermo Fisher Scientific (Saint-Laurent, QC, Canada).

### 2.2. Sample Preparation

The a-C:H:Ag and the a-C:H:AgO coatings were then prepared using a plasma enhanced chemical vapor deposition—physical vapor deposition hybrid reactor (PECVD-PVD, modified FLARION series system, Varennes, Plasmionique, QC, Canada). The preparation of the a-C:H coatings has been explained in detail elsewhere [5]. Briefly, it consisted of a continuous process in two successive sequences: H_2_ activation and a-C:H deposition using a mixture of CH_4_/Ar (1:1 ratio). The Ag and the AgO NPs, as dopant agents, were simultaneously introduced during a-C:H deposition using an RF-magnetron sputtering system with a commercially purchased Ag target (99.99%, Kurt J. Lesker Company, Jefferson Hills, PA, USA) and a plasma modified AgO target. The amount of Ag and AgO NPs deposited on the carbon matrix was controlled by the voltage applied on the Ag and the AgO target in the PECVD-PVD system. Subsequently, the a-C:H samples were introduced into a tubular Pyrex radio-frequency glow discharge (RFGD) plasma reactor (13.56 MHz, internal diameter: 1.9 cm) capacitively coupled through an impedance matching network. The reactor has been previously described in detail [28]. The deposition of the topmost polymeric coating was then performed using pulsed afterglow plasma polymerization at 11 cm below the discharge, with a mixture of C_2_F_6_ and H_2_, and using the following conditions (as summarized in Table 1): peak power input of 150 W, varying duty cycles (5.3%, 10%, and 20%), gas pressure of 700 mTorr, gas flow rate of 20 sccm (94% C_2_F_6_ and 6% H_2_), and deposition time of 5 min. These parameters have been selected according to previous studies; however, the duty cycle (DC) was varied to study its effect on surface porosity. Three different duty cycles were chosen to compare and assess the deposition the fluoropolymer (CF_x_) coating on a-C:H, a-C:H:Ag, and a-C:H:AgO, namely 5.3% (5.3CF_x_), 10% (10CF_x_), and 20% (20CF_x_). After plasma polymerization, the samples were removed from the reactor and stored under vacuum until further use. 

### 2.3. Surface Characterization

#### 2.3.1. X-ray Photoelectron Spectroscopy (XPS)

The chemical composition was assessed by X-ray photoelectron spectroscopy (XPS) and Auger electron spectroscopy (AES) using PHI 5600-ci equipment (Physical Electronics, Chanhassen, MN, USA). XPS and AES spectra were acquired at a detection angle of 45°, using a Kα line of a standard aluminum X-ray source operated at 300 W with a pass energy of 187.85 eV for survey and 5.85 eV for high resolution and Auger transitions. The curve fittings for high-resolution XPS spectra were determined by means of the least-squares method using Gauss–Lorentz functions with a Shirley background subtraction. All peak positions were normalized to that of the C1s peak, which was considered at 285 eV. The chemical form of silver has been analyzed using the modified Auger parameter (α’). The modified Auger parameter of silver was calculated by adding the binding energy of the Ag3d_5/2_ peak and the kinetic energy of the M_4_N_45_N_45_ Auger peak, as indicated on Equation (1): (α’ = *E_k_* (M_4_N_45_N_45_) + *E_b_* (3d^5/2^))(1)

Each condition was analyzed using one sample scanned at three different positions to monitor the homogeneity of the coatings.

#### 2.3.2. Contact Angle and Surface Energy 

Water contact angles (WCA) of the samples were measured by the sessile drop method with the VCA optima XE (AST Products, Billerica, MA, USA) using distilled water drops of 1 µL. For each condition, five drops were placed at different locations of one sample. The reported contact angles are the average of those measurements. Then, the surface energy of the samples was measured using the Fowkes Theory [29]. It describes the surface energy of a solid as having two components: a dispersive component and a polar component. Diiodomethane (50.8 mN/m) and water (46.4 mN/m) were used as probe liquids placing 5 drops of 1 µL at different locations of one sample per condition. 

#### 2.3.3. Atomic Force Microscopy (AFM)

Surface images were obtained using a Digital Instruments Dimension TM3100 atomic force microscopy (AFM, Santa Rosa, CA, USA) operating in tapping mode and equipped with an etched silicon tip (model NCHV, tip radius = 10 nm, Bruker). The roughness of the surface was evaluated using the root mean square roughness (R_RMS_) since it represents the standard deviation of the distribution of heights, and it is also more sensible to large deviation from the mean line [30]. The measurements were evaluated over 5 μm × 5 μm on one sample for each condition. 

#### 2.3.4. Scanning Electron Microscopy (SEM)

Surface morphology of the samples after contact with bacteria was analyzed using scanning electron microscopy (Quanta 250, FEI Company Inc. Thermo-Fisher Scientific, Hillsboro, OR, USA). Prior to analysis, the samples were fixed, dehydrated, and coated with a thin gold film to obtain scanning electron images with improved quality. Afterwards, secondary electrons (SE) were used to obtain at least three images of two samples per condition operated with an acceleration voltage of 15 kV. 

#### 2.3.5. Time-of-Flight Secondary Ion Mass Spectrometry (ToF-SIMS) 

All ToF-SIMS analyses were performed using a ToF-SIMS IV spectrometer (ION-TOF GmbH, Münster, Germany). A commercial PTFE sample (50 μm thickness, Goodfellow Ltd., Pittsburgh, PA, USA) was first analyzed to provide a reference material for evaluating the mass fragments of the fluoropolymer. ToF-SIMS spectra were acquired from 0 to 800 *m*/*z*. Elemental and molecular distribution maps, called images, were obtained using a 25 keV Bi_3_ = ion beam (current = 0.06 pA) rastered over an area of 100 μm × 100 μm with a pixel density of 256 × 256 pixels, in burst alignment mode (high lateral resolution mode) using one sample per condition.  A low-energy electron flood gun was used to ensure charge compensation. Depth profile measurements were obtained with a dual beam design, using 25 keV Bi_3_^+^ ions as primary beam (current = 0.25 pA, rastered = 150 μm × 150 μm, pixel density = 128 × 128 pixels, in high-current bunch mode (HCB)) and 500 eV C_s_^+^ ions as sputter beam (current = 30 nA; raster = 450 μm × 450 μm) using one sample per condition. The sputtering was operated in non-interlaced mode, with 5 s sputter time and 1 s pause. Each data point was acquired using 3 analysis frames. Similarly, a low-energy electron flood gun was used to ensure charge compensation. The profiling time was adapted for each sample and stopped when the signal of the substrate was apparent. The sputtering time needed to completely etch the coating depends on the thickness of each layer but also on the type of sample leading to variations in their sputtering yields. During imaging and depth profiling, the negative and positive polarity were analyzed; however, the negative polarity was chosen since the use of Bi_3_^+^ and Cs^+^ strongly favors the formation of negative ions [31].

### 2.4. Release Profile 

Release analysis from the samples was analyzed and performed in static conditions using DI millipore water. Three coated samples per condition were immersed in 2 mL of DI water and kept at room temperature for a period varying between 30 min and 7 days. The fluids were then sampled and analyzed by microwave plasma—atomic emission spectroscopy (MP-AES model 4100, Agilent Technologies, Santa Clara, CA, USA) with a detection limit of 1 µg/L. 

### 2.5. Antibacterial Essay

#### 2.5.1. Bacterial Strain and Culture Preparation

*E. coli* (ATCC 25922) was used in this study. The strain was maintained in TSB with 10% glycerol and stored at −80 °C. Prior to the experiment, the stock culture was streaked on Tryptic Soy Agar (TSA) and incubated at 37 °C for 24 h. Afterward, the bacteria were suspended in TSB medium at a concentration of 10^6^ cell/mL.

#### 2.5.2. Antibacterial Activity Test

The antibacterial behavior was studied using a modified version of the American Society for Testing and Materials International E2149 (ASTM E2149). In this case, the samples had a reduced surface area and longer contact with bacteria compared with the standard. The substrates were first sterilized using UV light for 30 min, flipping every 15 min. Next, 10 mL of the bacterial suspension was added to each substrate in a 50 mL tube and shacked incubated at 125 rpm for 12 h at 37°. Then, 100 µL serial tenfold dilutions of the resulting suspension were used to determine the number of colony-forming units (CFU). The results represent the average of at least three different samples for each condition. It is worth mentioning that after 12 h, a-C:H and CF_x_/a-C:H samples presented delamination and are not shown in the CFU results. 

After the assay was performed, the Ag-based samples were fixed and dehydrated to be analyzed by SEM. The process involved fixing the samples with a solution of 1% glutaraldehyde for 30 min at room temperature. Then, they were rinsed with deionized water for 5 min, repeating three times. Finally, the samples were dehydrated using four ethanol solutions with increasing concentrations (20%, 50%, 90%, and 100%) for 5 min, repeating two times for each one. The samples were removed from the solution and keep at room temperature until their use.

#### 2.5.3. Statistical Analysis 

Statistical significance was calculated using analysis of variance (ANOVA) non-parametric Kruskal–Wallis method with Dunn post hoc test through the software InStat™ version 3.05 (GraphPad Software, San Diego, CA, USA). Values of *p* < 0.05 or less were considered significant.

## 3. Results

### 3.1. Chemical Characterization

The survey results, presented in Figure 1a, show the atomic concentration for a-C:H films with and without the addition of the CF_x_ film. 

For the bare a-C:H coatings, survey analyses show an expected composition of mainly carbon and oxygen. The presence of silver is evidenced with an atomic concentration of 12.1 ± 0.2 at. % for a-C:H:Ag and 11 ± 3 at. % for a-C:H:AgO. To corroborate the chemical state of the silver, the modified Auger parameter (α_Ag_’, see Equation (1)) was evaluated (Appendix A). The α_Ag_’ values for metallic silver, disilver oxide (Ag_2_O), and silver oxide (AgO) are reported at 726.5 eV, 723.9 eV, and 724.4 eV, respectively [32]. Indeed, the obtained α_Ag_’ value corroborates that the oxidation state of the silver in a-C:H:AgO (724.8 ± 0.5 eV) corresponds to that of silver oxide, as expected. In the case of a-C:H:Ag (725.5 ± 0.2 eV), the resulting value suggests a mixture between mostly metallic and partially oxidized silver [5,33].

The incorporation of a fluoropolymer (CF_x_) coating is evidenced by the presence of fluorine regardless of the duty cycle and the a-C:H surface used (Figure 1a). However, as observed in Figure 1b, decreasing the pulse off time changes the composition of the fluoropolymer film considerably. Indeed, for the CF_x_/a-C:H coating, the highest F/C ratio is achieved using 5.3% DC; this value then decreases below 1 as a shorter pulse off time is used. Regarding the coatings containing Ag and AgO NPs, a different behavior is observed. For the CF_x_/a-C:H:Ag and CF_x_/a-C:H:AgO coatings, the highest F/C ratio was obtained at 5.3% DC and 20% DC, respectively. Interestingly, both coatings displayed the lowest F/C value using 10% DC, which is accompanied with a slight increase in the silver amount detected. Nevertheless, it is important to mention that a concentration between 3 and 8 at. % of silver was detected regardless of the duty cycle used (Figure 1a). 

The influence of the duty cycle on the chemical composition of the a-C:H films has been further investigated with high-resolution XPS. The C1s spectra of the CF_x_ films (Table 2) were decomposed into five peaks, assigned to C-H and C/C (BE = 285–285.8 eV), -C-CF (BE = 286.5–287.8 eV), -CF (BE = 288.6–290 eV), -CF_2_ (BE = 292 eV), and -CF_3_ (BE = 294–294.8 eV) [24]. As oxygen was detected in the survey spectra, the bands at 286.4–287 eV, 288–288.4 eV, and 289.1–289.6 eV can also be attributed to oxygen groups such as C-O, C=O, and O-C=O, respectively. The components of interest are C-C/C-H, CF_2,_ and CF_3_, as they are directly related to chain organization. Indeed, CF_x_ are characteristic of chain length and CF_3_ of chain termination [34]. The undoped CF_x_/a-C:H sample exhibited an increase in the CF_2_ and CF_3_ content as the pulse off time was increased (5.3% DC). Intriguingly, for CF_x_/a-C:H:Ag, the highest CF_x_ content was observed when using 5.3% and 20% DC. Whereas for CF_x_/a-C:H:AgO, a higher content of CF_2_ and CF_3_ was observed by decreasing the pulse off time (20% DC), agreeing with the F/C ratios. 

Regarding the surface wettability of the films, uncoated a-C:H, a-C:H:Ag, and a-C:H:AgO showed contact angles of 90.4 ± 0.5°, 92 ± 4°, and 73 ± 1°, respectively. The slight decrease in the contact angle observed for the AgO-doped coating is correlated to the oxidation state of the NPs, as observed previously, and suggests a more hydrophilic surface. Then, the total surface energy for a-C:H, a-C:H:Ag, and a-C:H:AgO was estimated as the sum of the dispersive and the polar components which resulted in 40.3 ± 0.7 mN/m, 49.2 ± 0.7 mN/m, and 39 ± 2 mN/m, respectively. These results are in accordance with previous findings, where the incorporation of metallic NPs increases the total surface energy of the a-C:H films [5]. However, the further deposition of the fluoropolymeric film as the top layer lowered the total surface energy, leading to more hydrophobic surfaces, as presented in Figure 2. When using 5.3%, 10%, and 20% DC, the reported WCA values increased to: 118.6 ± 0.3°, 122.8 ± 0.4°, and 120.0 ± 0.5° for CF_x_/a-C:H films; 120.0 ± 0.6°, 117 ± 1°, and 119.7 ± 0.3° for CF_x_/a-C:H:Ag films; and 118 ± 5°, 117 ± 1°, and 115 ± 3° for CF_x_/a-C:H:AgO films, respectively. Their respective WCA images are shown in Appendix A. Regarding the total surface energy, when using the CF_x_ layer, the values considerably decreased and ranged between 10.9 and 8.3 mN/m, 12.5 and 9.5 mN/m, and 13 and 10.5 mN/m for CF_x_/a-C:H, CF_x_/a-C:H:Ag, and CF_x_/a-C:H:AgO films, respectively. 

The fluoropolymer coatings show significant variations in composition and surface energy, which can be correlated to the varied duty cycles, but they also depend on the underlying surface chemistry. In fact, these variations were more important for the coatings with added Ag and AgO NPs, meaning that the growth of the fluoropolymer is impacted. Therefore, the surface morphology of the modified coatings has been thoroughly investigated by AFM and is shown in Figure 3. 

### 3.2. Surface Morphology

Analysis of the images in Figure 3 shows the variation of a non-uniform coating with the presence of nanometric holes and ribbon-like features on the surface of the modified films depending on the surface chemistry underneath and the duty cycle applied. Nonetheless, the images show a smooth surface in a nanometric scale varying between 0.9 nm and 1.8 nm. 

The presence of holes is visible on all the surfaces, and their diameters are measured. It clearly appears that their sizes are dependent on the DC used. In fact, with 5.3% DC, the hole diameter varied between 118 and 431 nm for CF_x_/a-C:H, 157–314 nm for CF_x_/a-C:H:Ag, and 78–176 nm for CF_x_/a-C:H:AgO. When increasing the DC, their diameter increased between 275 and 863 nm with 10% DC and 353 and 706 nm with 20% DC for a-C:H; from 333 to 490 nm with 10% DC and to 392 nm with 20% DC for CF_x_/a-C:H:Ag; and between 176 and 529 nm with 10% DC and 196 and 294 nm with 20% DC for CF_x_/a-C:H:AgO samples. However, their presence is more apparent when using a DC of 5.3%, whereas their occurrence decreases as the DC is increased up to 20%. Indeed, only one or two holes were found on the surface when using 20% DC, suggesting a more homogenous deposition.

In addition to nanometric holes, the occurrence of ribbon-like features is evidenced when using 10% DC on the surface of CF_x_/a-C:H and on a CF_x_/a-C:H:Ag sample, both several micrometers long, whereas only small (with a lower spread in height) and randomly scattered particles (brighter spots) are present on the CF_x_/a-C:H:AgO surface. The presence of holes and ribbon-like features has been observed previously in several studies during plasma deposition of fluoropolymers, and can be attributed to the film growth mechanism, where initial nucleation sites (brighter spots) are formed, which then grow and multiply to create ribbon-like features, thereby increasing the surface roughness [34]. A different behavior is perceived in the case of CF_x_/a-C:H:AgO, where higher nucleation density (brighter spots) and roughness are also observed when using 5.3% DC. By increasing the duty cycle, the nucleation density then slightly decreases, without the presence of ribbon-like features whatsoever. 

In light of the obtained results, 5.3% DC was chosen to further investigate the overall effect of depositing an additional topmost polymeric layer with the objective to elongate the release of Ag^+^ and provide an additional antibacterial mechanism. Indeed, a smaller pore size has been shown to lengthen the concentration of Ag^+^ released, whereas a relatively high F/C ratio and a low surface energy have shown to enhance the antibacterial effect of fluoropolymer films [22,26]. Additional analyses like ToF-SIMS images and depth profile, Ag^+^ release profile, and antibacterial tests were therefore performed on CF_x_/a-C:H, CF_x_/a-C:H:Ag, and CF_x_/a-C:H:AgO samples produced using 5.3% DC.

### 3.3. ToF-SIMS Analysis

ToF-SIMS imaging and depth profiling were used to better understand the interaction between the fluoropolymer (at 5.3% DC) and the modified a-C:H coatings with Ag and AgO NPs. Indeed, the main interest when using ToF-SIMS was to study the lateral homogeneity and the depth distribution of the 5.3CF_x_ film on the a-C:H coatings, which can help predict their permeability. Indeed, as previously stated, the addition of a fluoropolymer layer (highly hydrophobic) is expected to limit bacteria adhesion, while the defects or holes will allow the Ag^+^ to be released. For this purpose, the F^−^ ion was identified to represent the fluoropolymer layer, while the C_4_H^−^ ion was used to represent the a-C:H matrix, for both static and depth profiling conditions. In addition, SiO_2_^−^ and Ag^−^ ions were designated to represent, respectively, the silicon substrate and the distribution of silver NPs (Ag and AgO NPs) throughout the coating. 

Regarding the imaging of the fluoropolymer coating, the F^−^ ion was identified on all the surfaces, as shown in Figure 4. Amongst all, the 5.3CF_x_/a-C:H:AgO surface shows the most homogenous distribution of F^−^ concentration. However, after careful examination, small (roughly 1 µm in size) and scattered spots are detected, indicating a decrease in F^−^ intensity. Similar observations were made on 5.3CF_x_/a-C:H and 5.3CF_x_/a-C:H:Ag images, the latter exhibiting the lowest density of F^−^ concentration over the entire surface. Moreover, low intensities of C_4_H^−^ (attributed to a-C:H coatings) are observed on all the CF_x_ films, which could be related to the presence of holes in a nanometric scale evidencing the substrate underneath.

ToF-SIMS depth profile analysis of 5.3CF_x_/a-C:H, 5.3CF_x_/a-C:H:Ag, and 5.3CF_x_/a-C:H:AgO is shown in Figure 5. The ToF-SIMS technique is not quantitative in nature, and since no absolute calibration concerning sputtering depth and ion concentration has been performed, only qualitative profiles are obtained. Figure 5 shows the sputtering time required to pass through each layer starting from the CF_x_ film (represented by F^−^), then a-C:H matrix (represented by C_4_H^−^) unloaded and loaded with Ag and AgO NPs (107Ag^−^ and 109Ag^−^ isotopes), down to the silicon substrate (characterized by SiO_2_^−^). It is important to note that both Ag isotopes are shown in the depth profiling to represent the presence of Ag. Specifically, Ag is present in the coating when the intensity signals of both isotopes overlap since they have a mass ratio of approximately 1.0. For simplicity purposes, the 107Ag^−^ and 109Ag^−^ fragment are defined as Ag^−^ in the following description and in this manuscript. That being said, it takes approximately 35 s, 50 s, and 100 s for sputtering F^−^, and 650 s and 325 s to sputter Ag^−^ from 5.3CF_x_/a-C:H, 5.3CF_x_/a-C:H:Ag, and 5.3CF_x_/a-C:H:AgO, respectively. Although the same deposition parameters were used for each layer, it is evident that different sputtering times are obtained. This can be attributed to different growing mechanisms of the CF_x_ layer depending on the type of underlying surface, as evidenced by AFM analyses. Regarding the Ag^−^ signal, different behaviors are observed depending on the deposition process involving the incorporation of Ag or AgO NPs. For the 5.3CF_x_/a-C:H coating containing Ag NPs, the Ag^−^ signal can be divided in three different zones. First, over the initial 50 s, the intensities of signals F^−^ and Ag^−^ do not reach their maximum at the same sputtering time. This suggests a higher concentration of silver between the CF_x_ and the a-C:H interface (0–200 s); thus, a strong mixing between silver and the fluoropolymer can be expected. Second, the Ag^−^ remains constant in the upper part of a-C:H (200–500 s), suggesting a homogenous distribution of the NPs in the matrix. Third, in the bottom part of a-C:H, the Ag^−^ signal drops to noise level. This is in accordance with the deposition process of a-C:H:Ag, where Ag NPs are incorporated after the a-C:H matrix was grown on the Si substrate. In the case of the 5.3CF_x_/a-C:H coating containing AgO NPs, the same three different zones are observed. However, two main differences are visible in the profile; the transition between the first two zones is less abrupt and the Ag^−^ signal in the second zone is spread over a narrower sputtering time interval (no plateau as previously observed) before dropping to noise level. These results are in correlation with the sputtering time needed to pass through the a-C:H layer (1710 s, 1400 s, and 1035 s to sputter C_4_H^−^ from 5.3CF_x_/a-C:H, 5.3CF_x_/a-C:H:Ag, and 5.3CF_x_/a-C:H:AgO, respectively) and with previous findings where the deposition rate and the growing mechanism of the a-C:H film depends on the nature of the target used to sputter Ag or AgO NPs [5]. Specifically, the use of a metallic silver target results in more energetic condensation of Ag particles and a higher density of hydrocarbon molecules. In contrast with a silver oxide target, the presence of atomic oxygen radicals and O_2_^+^ species can promote carbon etching and decrease the generation of hydrocarbon ion densities, and consequently, a thinner a-C:H coating.

It is worth noting the higher intensities of Ag^−^, F^−^, and SiO_2_^−^ ions at the interface between two layers, either between the CF_x_ film and a-C:H matrix or the a-C:H matrix and Si substrate. These variations, attributed to the transition between two layers, have been attributed to a matrix effect [35,36,37,38]. Similarly, both Ag isotopes are present in the Si substrate region for all the conditions. However, the fact that both Ag isotopes (with a mass ratio of approximately 1.0, i.e., the intensities should overlap and not separate) appear on the depth profiling of 5.3CFx/a-C:H, when in theory, it should show the absence of any Ag fragments, evidences that the observed Ag fragments on 5.3CFx/a-C:H:Ag and for 5.3CFx/a-C:H:AgO samples (after approximately 1250 s and 1000 s, respectively) do not correspond to Ag from the coating, the target, or the deposition process.

All in all, ToF-SIMS analyses suggest the presence of Ag and AgO NPs in the 5.3CF_x_/a-C:H-based coatings’ interfaces, as desired. Thus, a continuous flow of Ag^+^ is expected to pass throughout the CF_x_ layer in a controlled manner. 

### 3.4. Release of Silver Ions

Figure 6a,b show the release profiles of Ag^+^ from a-C:H:Ag, a-C:H:AgO, 5.3CF_x_/a-C:H:Ag, and 5.3CF_x_/a-C:H:AgO samples immersed in deionized water for 30 days to highlight the effect of the fluoropolymer topmost layer deposited using 5.3% DC. 

Uncoated samples a-C:H:Ag and a-C:H:AgO were first analyzed and the results show a maximum Ag^+^ concentration of 0.44 ± 0.03 mg/L and 1.12 ± 0.09 mg/L after 30 days (Figure 6a), respectively. As already reported, oxidizing Ag NPs accelerates the release of Ag^+^ without increasing the loading concentration of NPs in the coating. This strategy avoids the use of an excessive amount of Ag NPs and efficiently releases Ag^+^. However, it is important to note that between 24 h and 14 days, the concentration of Ag^+^ stabilizes and slowly reaches a saturation point on both coatings. In this case, the introduction of an additional topmost fluoropolymer (Figure 6b) flattens the saturation curve, delays the release of Ag^+^ and prolongs the complete depletion of the NPs from the a-C:H matrix. After 30 days, the Ag^+^ concentration released from 5.3CF_x_/a-C:H:Ag and 5.3CF_x_/a-C:H:AgO reached 0.50 ± 0.09 mg/L and 0.493 ± 0.008 mg/L, respectively. Although this type of strategy decreases the concentration of Ag^+^ released in the first few hours, a slow continuous release (without reaching a saturation point) is particularly desired on environmental surfaces for longer efficacy. In this sense, the incorporation of the 5.3CF_x_ film not only plays a key role in tuning the release profile of Ag^+^ but may also function as an additional antibacterial mechanism due to its high hydrophobic nature. The latter aspect will be explored in detail in the next section.

### 3.5. Antibacterial Activity

The antibacterial results in Figure 7 demonstrate a significant decrease in *E. coli* growth after 12 h in direct contact with both a-C:H:Ag and a-C:H:AgO, in comparison with Si (negative control). The antibacterial activity of Ag NPs has been widely reported in literature and it has been well demonstrated to be dependent on Ag concentration in the coating; higher concentration of Ag NPs, higher antibacterial activity [39,40]. However, by incorporating AgO NPs in the coating (with the same concentration of Ag), it is possible to induce a slightly higher inhibition of bacterial growth (77% inhibition) in comparison with metallic Ag NPs (69% of inhibition). Regarding the addition of the CF_x_ layer, 5.3CF_x_/ a-C:H:Ag showed an inhibition of 88%, 19% higher than the uncoated a-C:H:Ag film. Evidently, the incorporation of a hydrophobic layer exerts an added antibacterial effect. As for the 5.3CF_x_/a-C:H:AgO coating, unexpectedly, no antibacterial activity was reported after 12 h. It is important to note that the antibacterial activity of Ag-based coatings not only depends on the amount of Ag NPs but also, in this case, on their ability to penetrate through the top CF_x_ layer and into the bacteria medium.

After 12 h in contact with *E. coli*, the samples were then examined by SEM. This analysis was used to qualitatively determine the number and the morphology of bacteria remaining on the surface for each condition. Figure 8 shows SEM images of *E. coli* on the surface of (i) Si, (ii) a-C:H:Ag, (iii) a-C:H:AgO, (iv) 5.3CF_x_/a-C:H:Ag, and (v) 5.3CF_x_/a-C:H:AgO films. Among all the conditions, a-C:H:AgO and 5.3CF_x_/a-C:H:Ag images display the presence of mostly single and isolated bacterium. On the other hand, the surface of 5.3CF_x_/a-C:H:AgO shows a high density of well-defined rod-shaped bacteria, as well as the attachment of aggregated and multilayered bacteria, which could indicate the initial stages of biofilm formation and is in accordance with results in Figure 7. Regarding sample a-C:H:Ag, the SEM image evidences the attachment of a mixture of monolayered and single bacteria on the surface. For the a-C:H:AgO sample, it is worth mentioning the appearance of craters/pits on the bacterial surface. Similarly, 5.3CF_x_/a-C:H:Ag displays the presence of only disrupted single bacterium on the surface. The occurrence of craters/pits on bacteria, when in contact with silver-based coating, is a well-known phenomenon reported in the literature and has been attributed to the disruption of bacterial cell walls by Ag^+^ [20,41]. In this sense, taking into consideration both quantitative and qualitative results, 5.3CF_x_/a-C:H:Ag successfully delays the formation of biofilm and promotes the inactivation of *E. coli* by the disruption of their cell wall after 12 h.

## 4. Discussion

In line with previous studies in the literature, it has been shown that it is possible to tune the composition of a fluoropolymer film deposited by low pressure plasma. To the authors’ best knowledge, this is the first study where the presence of a porous fluoropolymer with circular defects or holes is desired and used as a physical barrier to modulate the release of Ag^+^ from Ag-based a-C:H coatings. Indeed, the results obtained herein indicate the potential offered by variable duty cycle pulsed plasma polymerization in tailoring not only the surface composition but also the surface morphology, the release of Ag^+^, and consequently, the antibacterial activity. Therefore, the pertinence of this work falls on using the deposited topmost fluoropolymer layer to elongate the release of Ag^+^ whilst increasing the antibacterial activity of the coating (i.e., multifunctional or synergistic approach). These findings hold significant relevance when designing Ag-based antibacterial coatings, since bacteria are constantly adapting and changing. As a result, most studies have focused on solely increasing the concentration of Ag in the coating. However, higher concentrations of silver can lead to adverse health effects when in contact with human cells, especially on surfaces that are frequently touched by multiple individuals in healthcare settings (i.e., environmental surfaces). More importantly, bacteria may develop resistance to Ag over time if constantly exposed to high concentrations. Herein, the main focus is not only Ag concentration but, more importantly, on developing an efficient method to increase their antibacterial activity while minimizing any potential risks.

### 4.1. Effect of the Duty Cycle on the Deposition of CF_x_ Films

Understanding the deposition process of fluoropolymers is essential in controlling the structure and the surface morphology of the final CF_x_ coatings. The present modulated experiments have shown the influence of varying the duty cycle during pulse plasma polymerization for obtaining a hole-dense surface. Indeed, for all the CF_x_/a-C:H-based surfaces, longer pulse off times (i.e., shorter DC, 5.3%) increased the hole density, as shown in Figure 3. A plausible explanation can be attributed to the polymerization process itself. In pulse plasma polymerization, it is more probable to favor chain propagation during each pulse by attachment of the monomer molecules at one or few radical sites [42]. During the next pulse, at another radical, the chain propagation continues. Then, by increasing the pulse off time, the polymerization rates decrease [42]. Thus, with shorter DC, the final fluoropolymer consists of short segments of polymeric chains with more irregularities (e.g., circular defects). 

As evidenced by the XPS, AFM, and ToF-SIMS results, the presence of Ag and AgO NPs on the surface influenced the deposition of the CF_x_ layer. In general, it is evident that the hole density decreased and the F/C ratio remained under one regardless of the DC used, with the exception of 20% DC for CF_x_/a-C:H:AgO. Specifically for CF_x_/a-C:H:Ag coatings, a shorter pulse off time (i.e., longer DC, 20%) achieved less hole density and decreased CF_2_ content, whilst the CF_3_ proportion remained unchanged. In addition, the presence of Ag NPs caused tertiary and quaternary carbons (i.e., CF and C-CF), corresponding to branched structures, to increase. Similar results were obtained by Cioffi et al., where the incorporation of Au NPs on a Teflon-like film was related to a decrease in the intensities of the carbon species belonging to linear chains, such as CF_2_. Their changes were related to a restructuring of the polymeric matrix, undergoing cross-linking, thus becoming more organized and tied [43,44,45]. In the case of CF_x_/a-C:H:AgO coatings, an opposite trend is observed. By increasing the DC, the content of CF_2_ and CF_3_ species increases while the proportion of CF and C-CF bonds remains fairly similar. Typically, the presence of oxygen in the plasma gas tends to consume CF_2_ through the formation of COF, CO, CO_2_, and/or F, thus reducing the deposition rate and favoring the etching process rather than deposition [46]. However, the presence of oxygen during plasma polymerization has also shown to increase the formation of a higher density and smoother polymer [47]. In this case, the presence of an oxygen-rich surface (due to the presence of AgO NPs) enhanced the formation of carbon species related to linear chain and chain termination, when increasing the DC (Table 2). Specifically, the plasma polymerization using 20% DC on the oxygen-rich surface stabilized the forming film by creating a surface with fewer defects and lower hole density [48,49,50,51]. 

### 4.2. Use of 5.3% DC Fluoropolymeric Film as a Barrier Coating to Enhance Antibacterial Activity 

The interplay between surface morphology and surface energy, as well as the incorporation of silver NPs, a well-known bactericidal agent, is translated to significant differences in the antibacterial behavior of the tested surfaces. Herein, two different parameters were taken into consideration to improve the antibacterial activity of a-C:H-based coatings: the oxidation of Ag NPs and the incorporation of a fluoropolymer coating. As evidenced by the release profile (Figure 6a) and the antibacterial essays (Figure 7 and Figure 8), the use of AgO NPs in a-C:H coatings is a simple and efficient approach to activate the release of Ag^+^ and improve their antibacterial effect, without increasing the concentration of silver. Their effect on the a-C:H matrix and the mode of action have been extensively explained in a previous study [5]. However, as reported by Ellinas et al., there is a limit to the antibacterial action of any engineered surface [52]. In this case, the limit was reached when the highest concentration of AgO NPs (11 ± 3 at. % Ag) was used in these a-C:H coatings. Over the last decade, the integration of both antibacterial NPs and hydrophobic polymers has been demonstrated as one of the most efficient strategies to produce a coating with strong antibacterial activity [53]. In this study, a fluoropolymer layer with pores in a nanometric scale was deposited on a-C:H:Ag and a-C:H:AgO coatings to increase their antibacterial action and demonstrate a hybrid approach with both antiadhesive and bactericidal properties. For this purpose, 5.3CF_x_ was chosen and introduced to increase the WCA, render a lower surface energy, and control the release of Ag^+^, as shown in Figure 2 and Figure 6.

Unexpectedly, no antibacterial activity was reported for the 5.3CF_x_/a-C:H:AgO coating. There are different possible explanations to determine their null antibacterial activity at 12 h, such as a higher roughness value (i.e., 1.7 ± 0.5 nm, Figure 3), a smaller size range of holes (78–176 nm, Figure 3), a bimodal particle size distribution (Appendix A), and the incubation time of the samples with *E. coli*. Indeed, it has been reported that irregularities in polymeric surfaces promote bacterial adhesion and biofilm formation, unlike smooth surfaces that do not favor bacterial deposition [54]. This can be attributed to a greater surface area and the fact that depressions in roughened surfaces provide more favorable sites for colonization. It is also worth mentioning the difference in the average particle size and distribution between NPs on a-C:H:Ag and a-C:H:AgO coatings (Appendix A). While Ag NPs exhibit an average particle size of 27 ± 5 nm and a normal distribution, the histogram of AgO NPs yields an average particle size of 23 ± 5 nm and a bimodal size distribution, with two different modes at 20 nm and 33 nm. In combination with a smaller size range of holes, these factors could limit the penetration of bigger (i.e., >30 nm) AgO NPs [53]. In this context, it is possible that after 12 h, 5.3CF_x_/a-C:H:AgO had completely released the AgO NPs small enough to pass through the CF_x_ layer. Indeed, after 4 h in contact with *E. coli*, the 5.3CF_x_/a-C:H:AgO coating exhibited significantly higher bacterial inhibition with 63% in comparison with 46% inhibition for the 5.3CF_x_/a-C:H:Ag sample (Appendix A). Afterwards, a combination of all these factors mentioned beforehand come into play, weakening the antibacterial activity of the coating after 12 h.

For 5.3CF_x_/a-C:H:Ag, the incorporation of a CF_x_ coating successfully delayed bacteria growth and biofilm formation, whereas Ag NPs efficiently inactivated and disrupted bacteria cell walls. Even though this coating released a lower concentration of Ag^+^ for the first 24 h (0.16 ± 0.04 mg/L), in comparison with uncoated a-C:H:Ag (0.41 ± 0.02 mg/L), it is evident that the release profile determinates how Ag^+^ will interact with bacteria, where the CF_x_ layer allows for a slow and continuous release of Ag^+^. Indeed, using this hybrid scenario, the required metal quantity is significantly reduced, diminishing any potential toxic effect of silver [52]. Similar results were obtained by Yin et al., where the addition of a hydrophobic layer prolongated the release of Ag^+^, exhibiting antibacterial activity until 14 days [22]. Their obtained results were compared to a traditional silver-based coating with a higher released concentration of Ag^+^ and lower antibacterial activity at that time point. In this context, the use of a hydrophobic coating enhances the stability of silver-based coatings and prolongs the duration of the sustained release of antibacterial Ag^+^. By combining these two different approaches, it is possible to provide longer and thorough protection against bacterial colonization and transmission on environmental surfaces in healthcare settings.

## 5. Conclusions

In this study, an additional fluoropolymer layer produced by low pressure plasma was deposited on a-C:H films containing Ag (12.1 ± 0.2 at. % of Ag) and AgO NPs (11 ± 3 at. % of Ag). The main objective was to study the influence of CF_x_ on the release profile of Ag^+^ and the overall antibacterial activity of the coating. The deposition of the porous CF_x_ layer was explored by varying the duty cycle. In this sense, the incorporation of the CF_x_ layer produced at 5.3% DC proved to modify the release profile, by delaying the release of Ag^+^ and prolonging its complete depletion from the a-C:H matrix. Even though the released concentration decreased in comparison with the uncoated a-C:H:Ag and a-C:H:AgO films, it is evident that the release profile determinates how Ag^+^ will interact with bacteria, where the 5.3CF_x_ layer allowed a rather slow and sustained release without reaching a saturation point. Indeed, out of all the conditions, 5.3CF_x_/a-C:H:Ag successfully delayed the formation of biofilm and promoted the inactivation of bacteria by the disruption of their cell walls. Thus, using a hybrid approach composed of Ag NPs and a hydrophobic polymeric layer, it was possible to increase the overall antibacterial activity of the coating. The impact of this research highlights the importance of using multi-approach coatings as a promising tactic to overcome the inherent challenges associated with each strategy and as multiple lines of defense against bacteria on environmental surfaces to help prevent and limit the spread of HAIs.

Further studies are needed to better understand and improve the performance of CF_x_/a-C:H:AgO coatings, with special attention to the polymerization parameters used to deposit the additional CF_x_ layer. In addition, further experimental investigations are needed to perform an exhaustive antibacterial assessment with longer contact time points and including Gram-positive bacteria.

## Figures and Tables

**Figure 1 nanomaterials-14-00609-f001:**
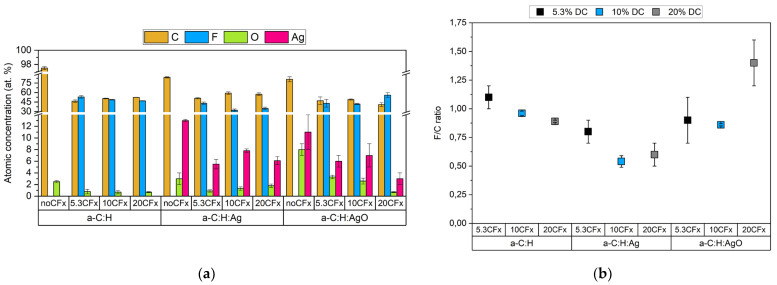
Chemical composition of the uncoated and CF_x_-coated a-C:H samples with different studied conditions. (**a**) XPS survey results and (**b**) F/C ratio per condition.

**Figure 2 nanomaterials-14-00609-f002:**
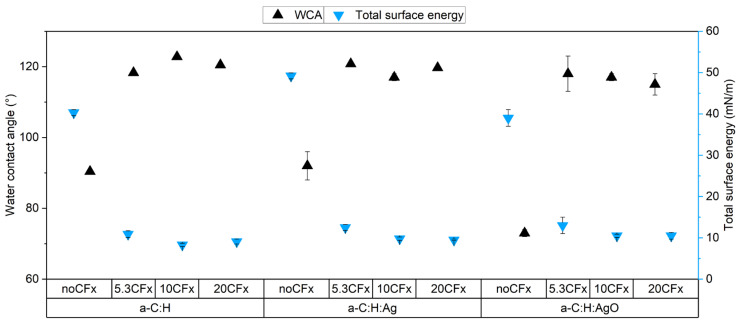
Summary of water contact angles (WCA) and calculated total surface energy of uncoated and CF_x_ a-C:H samples by studied duty cycle.

**Figure 3 nanomaterials-14-00609-f003:**
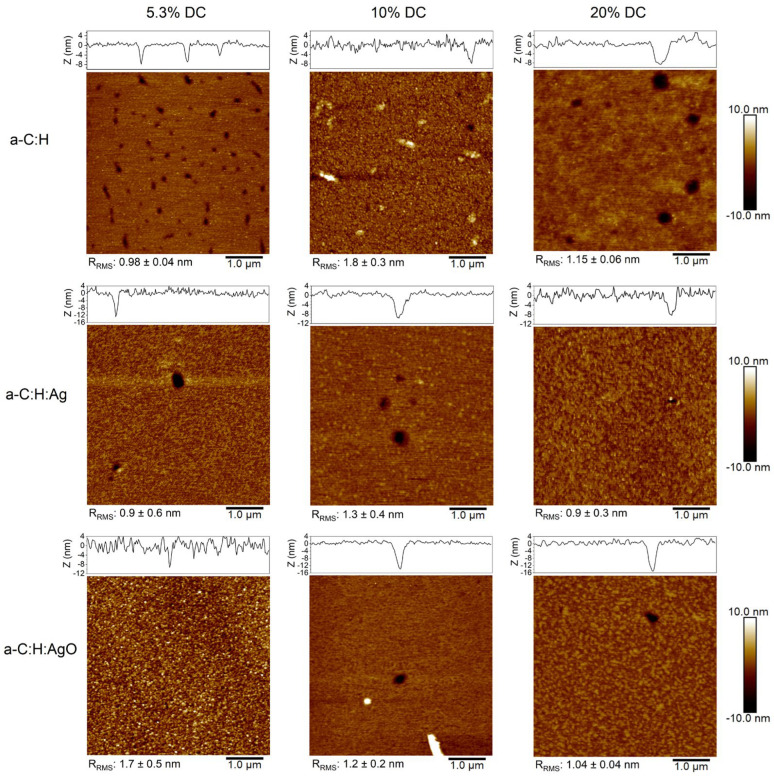
AFM images of (5 µm × 5 µm, in height mode) of the CF_x_-coated a-C:H samples depending on the duty cycle used and their respective roughness values.

**Figure 4 nanomaterials-14-00609-f004:**
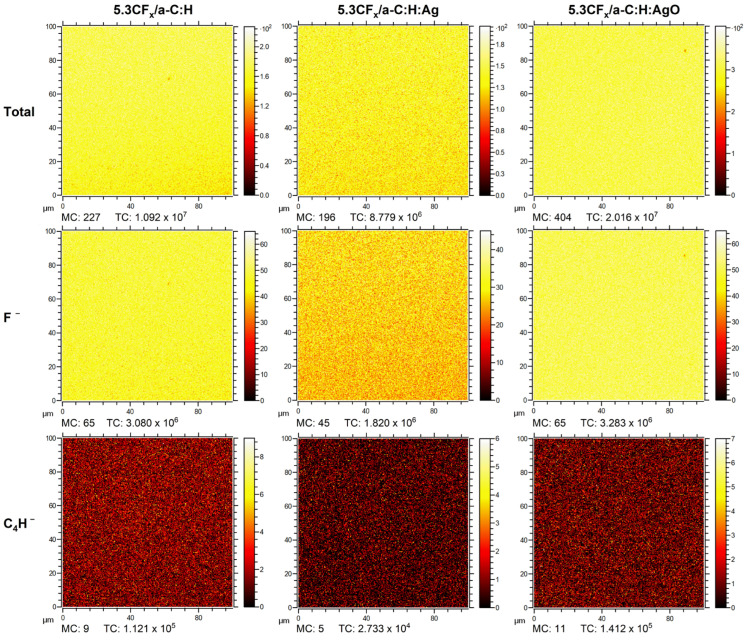
ToF-SIMS imaging mode of CF_x_/a-C:H, CF_x_/a-C:H:Ag, and CF_x_/a-C:H:AgO sing 5.3% DC with F^−^ and C_4_H^−^ chosen as specific fragments representing the CF_x_ layer and the a-C:H matrix, respectively.

**Figure 5 nanomaterials-14-00609-f005:**
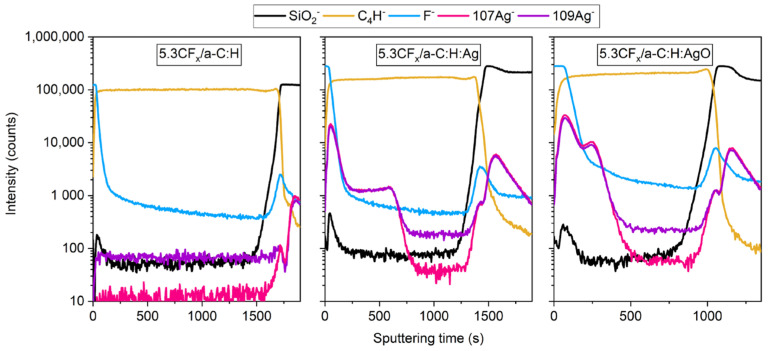
ToF-SIMS depth profile of 5.3CF_x_/a-C:H, 5.3CF_x_/a-C:H:Ag, and 5.3CF_x_/a-C:H:AgO showing F^−^, C_4_H^−^, 107Ag^−^, 109Ag^−^, and SiO_2_^−^ fragments representative of the CF_x_ layer, the a-C:H matrix, Ag and AgO NPs, and the Si substrate.

**Figure 6 nanomaterials-14-00609-f006:**
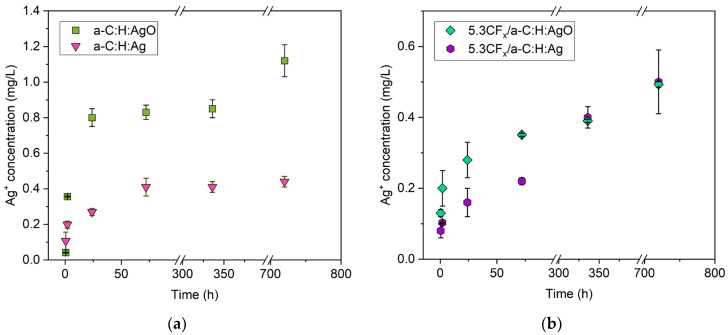
Release of silver ions (in mg/L) (**a**) before and (**b**) after CF_x_ deposition. The figures show the results of a-C:H:Ag (12.1 ± 0.2 at. % Ag), a-C:H:AgO (11 ± 3 at. % Ag), 5.3CF_x_/a-C:H:Ag, and 5.3CF_x_/a-C:H:AgO coatings in deionized water for 30 days. MP-AES measurements limits (not shown): limit of detection (LOD) at 1 µg/L and limit of quantification (LOQ) at 5 µg/L.

**Figure 7 nanomaterials-14-00609-f007:**
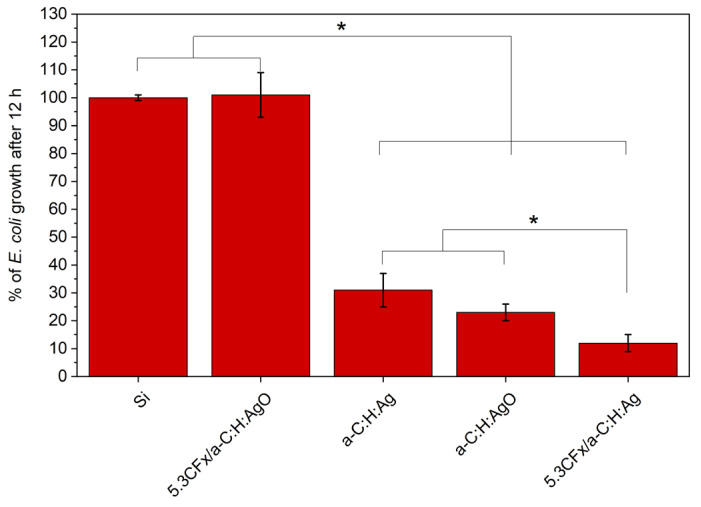
Percentage of *E. coli* growth after 12 h in contact with Si (as negative control) and modified coatings. The values were normalized to the Si substrate (* *p* < 0.05).

**Figure 8 nanomaterials-14-00609-f008:**
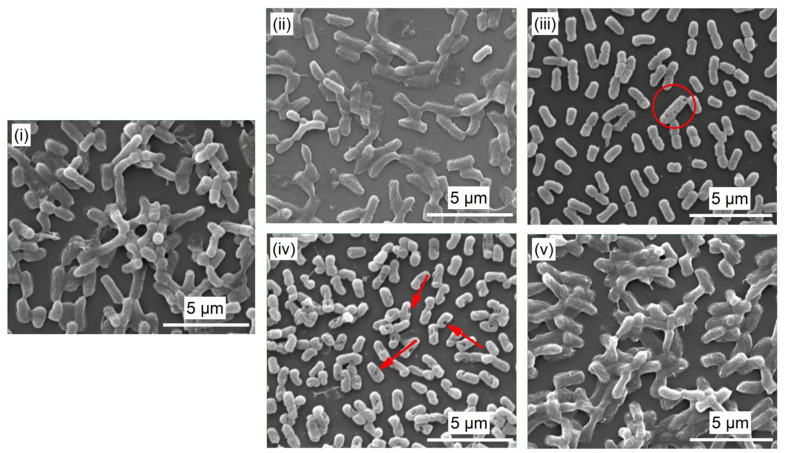
SEM surface images of (**i**) Si, (**ii**) a-C:H:Ag, (**iii**) a-C:H:AgO, (**iv**) 5.3CF_x_/a-C:H:Ag, and (**v**) 5.3CF_x_/a-C:H:AgO collected after antibacterial assays with an acceleration voltage of 15 kV and at an amplification of 10,000×. The red arrows and circle are used to emphasize the presence of a disrupted cell wall on the surface of bacteria.

**Table 1 nanomaterials-14-00609-t001:** Plasma deposition parameters for topmost fluoropolymer (CF_x_) coating.

Parameters	Polymerization
	C_2_F_6_ + H_2_
Peak power (W)	150
Duty cycles	5.3% (t_on_ = 5 ms, t_off_ = 90 ms)
	10% (t_on_ = 5 ms, t_off_ = 45 ms)
	20% (t_on_ = 5 ms, t_off_ = 20 ms)
Gas pressure (mTorr)	700
Gas flow rate (sccm)	19.0 for C_2_F_6_ and 1.2 for H_2_
Distance to the powered antenna (cm)	11
Treatment time (min)	5

**Table 2 nanomaterials-14-00609-t002:** Component proportions by high-resolution XPS of the fluoropolymer films deposited on a-C:H, a-C:H:Ag, and a-C:H:AgO depending on the duty cycle used (5.3%, 10%, and 20%).

Sample	Proportion (%)			
	Component	Duty Cycle		
		5.3%	10%	20%
CF_x_ a-C:H	C-C/C-H	32 ± 5	32 ± 1	39 ± 5
	C-CF	15 ± 5	19.0 ± 0.7	16 ± 3
	CF	8 ± 3	14.2 ± 0.4	13 ± 3
	CF_2_	35 ± 4	28 ± 1	26.0 ± 0.6
	CF_3_	9.6 ± 0.4	6 ± 1	5.8 ± 0.7
CF_x_ a-C:H:Ag	C-C/C-H	33 ± 2	47 ± 2	32 ± 8
	C-CF	22 ± 3	22 ± 7	28 ± 7
	CF	17 ± 2	15 ± 5	19 ± 1
	CF_2_	23 ± 4	12 ± 2	16.3 ± 0.6
	CF_3_	4.5 ± 0.8	4 ± 1	5 ± 1
CF_x_ a-C:H:AgO	C-C/C-H	41 ± 7	24 ± 7	14 ± 6
	C-CF	18 ± 3	28 ± 5	17 ± 5
	CF	11 ± 1	18 ± 2	16 ± 5
	CF_2_	22 ± 5	23 ± 2	43 ± 8
	CF_3_	8 ± 4	6 ± 3	10 ± 1

## Data Availability

The data presented in this study are available upon request to the corresponding author.

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
