# Peer review of "Fluorocarbon Plasma-Polymerized Layer Increases the Release Time of Silver Ions and the Antibacterial Activity of Silver-Based Coatings"

_nanomaterials, 2024, doi:10.3390/nano14070609_

Round 1

Reviewer 1 Report (Previous Reviewer 1)

Comments and Suggestions for Authors

The work of L. Bonilla-Gameros et al is devoted to the development of antibacterial coatings, which is an urgent task of modern materials science. The focus of this study is to modify the surface of coatings based on a-C:H and silver to increase hydrophobicity and change the release dynamics of the antibacterial component.

Unfortunately, the updated version of the manuscript still raises a number of fundamental questions. The main ones are still related to chemistry and research methodology. Therefore, I am forced to repeat the previous review, adding questions caused by changed points in the manuscript.

1) ! The authors do not prove the chemical form of silver in the obtained samples. This is unacceptable.

Moreover, I am surprised under the chemical formula “AgO”. It is repeated throughout the manuscript, so it is not a typo. Is it really silver(I,III)??? Then explain how the target was prepared and used, because this compound decomposes with the release of oxygen when heated (~100°C)!!

If silver oxide Ag2O was meant, then it is also unclear how it can be used in the method used (it also has low thermal stability).

In general, such chemical errors undermine the credibility of the authors.

In the updated version, the authors added a single mention of “The chemical form of silver has been analyzed using the modified Auger parameter (α’).” (line 119). This is not enough; evidence and primary data must be clearly provided. Moreover, it is unclear why the authors cannot determine the chemical form of silver using XPS data (line 112)?

2) The data provided (Supplemental Materials) shows that both Ag-containing materials are in the form of thin films rather than isolated nanoparticles (NPs). Thus, it is necessary to change the relevant terms throughout the manuscript or provide other evidence.

3) It is necessary to clearly determine the amount of silver introduced into all samples. For this, the authors can use MP-AES (line 173). It is important to prove quantitatively how comparable the molar content of silver in the studied samples is to each other. Further, it will be informative to provide graph of the Ag release as a fraction of the total content.

4) !! On lines 546-541, the authors indicate "thicker CFx layer" for sample 5.3CFx/a-C:H:AgO than for sample 5.3CFx/a-C:H:Ag. This does not allow comparison of results methodologically!

It is necessary to redo the samples with the same thickness of the top layer.

Then, authors need to specify the thickness of each layer for all samples under study. Moreover, it is necessary to specify the accuracy with which the layer thickness can be controlled.

5) For clarity, a flowchart of the sample preparation sequence should be provided. It is not clear why the authors chose exactly 5.3% DC, and not 5.0 and not 5.5 or 5.7%, etc. For the rest of the samples, there is no such accuracy (10% DC, not 10.3% DC). Explain the idea, please.

6) For clarity, the dynamics of silver release from samples with and without CFx layer should be presented on one graph.

7) !!! The results of antibacterial studies do not correlate with the dynamics of silver release and do not have a clear explanation. In fact, it is not clear why 5.3CFx/ a-C:H:Ag, which gives a low silver content in solution, exhibits the highest antibacterial activity. At the same time, the 5.3CFx/ a-C:H:AgO sample, which gives a similar dynamics of silver precipitation, does not show any activity at all. It turns out that the dynamics of the release of silver does not play a role at all??

The available explanations are not satisfactory. Apparently, there is a methodological error, indicated in Comment 4. Thus, a number of samples should be removed from the study and new ones should be added instead.

8) For clear comparison, antibacterial activity data must be provided for samples without silver, i.e. 5.3CFx/a-C:H and a-C:H. In the updated version, the authors added information that the fluoropolymeric films themselves also have an antibacterial effect (lines 310-311). Thus, it is necessary to understand whether the discussed effect (Figure 7) is due to the action of the 5.3CFx coating, or the silver (or a combination of both?). So far this is another methodological problem of the manuscript. Please note that the CFx coating thicknesses for 5.3CFx/a-C:H control samples must match those of 5.3CFx/a-C:H:Ag and 5.3CFx/a-C:H:AgO.

9) Regarding the discussion of the results, the authors noted that "Ag fragments on 5.3CFx/a-C:H:Ag and for 5.3CFx/a-C:H:AgO samples (after approximately 1250 s and 1000 s, respectively) do not orrespond to Ag from the coating or the deposition process." (lines 391-392). It is necessary to explain the reason for the appearance of these fragments. If these are matrix fragments, perhaps the calibration should be optimized? Otherwise, it is not yet clear how much we can trust this information. In fact, then the entire silver content is the sum of “intrinsic silver” (target, from the coating) and “matrix silver”. Authors should clearly indicate how the “intrinsic silver” can be separated, rather than leaving this problem to the readers to solve.

Comments on the Quality of English Language

Minor editing of English language required

Author Response

Reviewer 2 Report (Previous Reviewer 2)

Comments and Suggestions for Authors

As a revision, the authors have addressed the comments from the reviewers. The related data have been also provided in the revised manuscript. For example, the contact angle data. Therefore, I would like to recommend this manuscript to publish as its current form in Nanomaterials.

Author Response

Reviewer 3 Report (New Reviewer)

Comments and Suggestions for Authors

Overall the manuscript is ok. In the manuscript, the author has investigated the use of a 20 plasma fluoropolymer (CFx) as an additional top layer to elongate the release of Ag+ and increase 21 the antibacterial activity due to its high hydrophobic nature.

1.The author can add more information on the novelty of the current research in the introduction part.

2. Under materials and methods section, information on different types of chemicals used should be provided first before the sample preparation.

3. The author should characterize more, using the TEM, zeta potential analysis.

4.In SEM the author should also take the image of controls.

5. In Figure 7, it is not clear what are the inset in each images, need to be described clearly in the footnote.

6. Discussion part should be elaborated more with more discussion on the probable uses of the current research in the biomedical field.

7. Conclusion should include the future prospectives.

Comments on the Quality of English Language

Minor editing of English language required

Author Response

Reviewer 4 Report (New Reviewer)

Comments and Suggestions for Authors

The manuscript “Fluorocarbon plasma-polymerized layer increases the release time of Ag ions and the antibacterial activity of Ag-based coatings” by Bonilla-Gameros L. et al. can be accepted for publication after the authors will properly address all the raised queries (in the order they appear in the manuscript): 

1. Acronyms should not be used in the title.

2. Some specific results should be included in the Abstract.

3. Typically, a-C:H-type coatings have high residual stress. This can lead to easy film delamination. The authors should comment on this phenomenon with respect to their prepared samples, taking into consideration also that they did not mention the thickness of their deposited coatings.

4. What was the concentration of silver used by the authors for both type of coatings reported in this study? Moreover, how did they measure the silver concentration in the coatings? This info is important and has to be included in the main text.

5. The number of samples prepared for each investigation method should be clearly mentioned.

6. “150 W” (page 3, Table 1) should read as “150”.

7. “The chemical characterization ….. by XPS” (page 5, lines 203 to 204); “In addition to …. measurements” (page 6, lines 246 to 247); “To determine …… theory [29]” (page 6, lines 251 to 252)”; “The ability …… feature” (page 11, lines 398 to 399); “The antibacterial …..activity [39-40]” (page 12, lines 425 to 430) – these are not results! These phrases should be therefore moved to their corresponding section.

8. “diameter” should read “diameters” (page 7, line 281).

9. The authors should explain the reason why they chose not to represent the antibacterial activity of the samples both in their initial stages of interaction (i.e., T0 and a few hours later – besides the 4h results shown in Supplementary material), and after prolonged immersion times (24 – 72h). It is well known that these time intervals are very important for an overall perception over the efficacy of the samples in contact with the microbial slurries.

10. It is a documented fact that the symbol “ns” means “p > 0.05”; “*” means “p < 0.05; “**” means p < 0.01; etc. Therefore, the authors should correct the legend of Figure 6 accordingly.

11. The authors state that “Afterwards, a combination of all these factors mentioned beforehand come into play, weakening the antibacterial activity of the coating after 12h” (page 15, lines 565 to 567). They have to present a clear proof for this statement.

12. The concentration of the used silver should be clearly mentioned in the “Conclusions” section, when summarizing the results.

13. At least one possible application of these results should be clearly mentioned in the “Conclusions” section.

Comments on the Quality of English Language

 Minor editing of English language required

Round 2

Reviewer 1 Report (Previous Reviewer 1)

Comments and Suggestions for Authors

Unfortunately, the corrections made, as well as the responses to comments, are not sufficient for publication and do not address the fundamental problems of the manuscript.

1) The authors still claim that the material they obtain contains silver oxide (I,III), AgO. They appeal to the modified Auger parameter, although they indicate that “disilver oxide (Ag2O)... are reported at ... 724.4 eV [32]. Indeed, the obtained αAg' values corroborates that the oxidation state of the silver in a-C :H:AgO (724.8 ± 0.5 eV)..." . Thus, apparently, the materials contained silver(I) oxide, Ag2O, which for some unknown reason is named as a mixed oxide AgO. Moreover, there is no evidence of the formation of a silver oxide (I,III) target under the conditions they described. This, to a certain extent, kills confidence in the results of the work.

2) The reviewer cannot accept that Figure S3 reflects individual silver nanoparticles. There are no significant gaps between grains in AFM images. Therefore, these are continuous films formed by grains of the specified sizes. The authors added Figure R1 for proof, but by comparing Figures S3 and R1, anyone can conclude that they appear to correspond to different materials. Thus, the reviewer begins to see this as data fraud.

3) The reviewer cannot agree that quantitative total silver content is a factor that can be omitted when developing an appropriate antibacterial material. This, along with the nature of the release, is one of the key factors, because is responsible for the duration of the effect.

Thus, it is necessary to clearly determine the amount of silver introduced into all samples. For this, the authors can use MP-AES (line 173). It is important to prove quantitatively how comparable the molar content of silver in the studied samples is to each other. Further, it will be informative to provide graph of the Ag release as a fraction of the total content.

4) The reviewer understands that the introduction of different chemical components may affect the speed of the fluoropolymer layer application process. However, from a scientific point of view, if the authors show "influence of the plasma-deposited fluoropolymer layer (CFx)", then the thicknesses of all these layers on all studied objects must be the same. Otherwise, this information indicates 2 independent parameters (1) the presence of a layer (2) the thickness of the layer. This is a gross methodological error. It is necessary to redo the samples with the same thickness of the top layer. It is necessary to change the process conditions (for example, time) so as to obtain layers of the same thickness. This uniformity must be proven, otherwise the current manuscript cannot be accepted.

5) Although quantitative antibacterial effects of silver-free materials, 5.3CFx/a-C:H and a-C:H, are key control experiments, the authors do not present these data, citing "delamination." However, in their previous work, relevant data for the a-C:H system was shown https://www.sciencedirect.com/science/article/pii/S0925963522006124.  Moreover, it is unclear why (a) the delamination effect was not observed during silver release studies (b) how the introduced nanoparticles prevent delamination so effectively. It is necessary to clearly discuss these points in the text of the manuscript.

6) The authors did not add sufficient and convincing explanations to the text of the manuscript regarding the following comment "!!! The results of antibacterial studies do not correlate with the dynamics of silver release and do not have a clear explanation. In fact, it is not clear why 5.3CFx/ a-C:H:Ag, which gives a low silver content in solution, exhibits the highest antibacterial activity. At the same time, the 5.3CFx/ a-C:H:AgO sample, which gives a similar dynamics of silver precipitation, does not show any activity at all. It turns out that the dynamics of the release of silver does not play a role at all?? The available explanations are not satisfactory. Apparently, there is a methodological error, indicated in Comment 4. Thus, a number of samples should be removed from the study and new ones should be added instead."

Comments on the Quality of English Language

Minor editing of English language required

Author Response

Please the attachment. 

Reviewer 3 Report (New Reviewer)

Comments and Suggestions for Authors

The authors have revised the manuscript as suggested.

Comments on the Quality of English Language

looks ok

Author Response

The authors would like to thank the reviewer for their time invested in the revision of this manuscript, for the relevance of the comments, and the suggestions to improve the quality of this work.

Reviewer 4 Report (New Reviewer)

Comments and Suggestions for Authors

The authors responded to all the raised queries.

Even though they stated that they understood and acknowledged my third comment (related to a-C:H films’ delamination), they mentioned (eight query, Reviewer 1) that “a-C:H and 5.3CFx/a-C:H have been disregarded since both coatings presented delamination during the analysis”. They should clearly mention this aspect in the revised version of the manuscript.

In the future, the authors should take care that after using the Word’s option “Track changes”, the number of rows will be changed when saving the document in pdf format. Therefore, it was quite difficult for this Reviewer to closely observe all performed modifications (considering the number of the rows indicated by the authors in their letter of response).

Comments on the Quality of English Language

Minor editing of English language required.

Author Response

The authors would like to thank the reviewer for their time invested in the revision of this manuscript, for the relevance of the comments, and the suggestions to improve the quality of this work.

This manuscript is a resubmission of an earlier submission. The following is a list of the peer review reports and author responses from that submission.

Round 1

Reviewer 1 Report

Comments and Suggestions for Authors

The work of L. Bonilla-Gameros et al is devoted to the development of antibacterial coatings, which is an urgent task of modern materials science. The focus of this study is to modify the surface of coatings based on a-C:H and silver to increase hydrophobicity and change the release dynamics of the antibacterial component. The results may be of interest to readers of Nanomaterials, but unfortunately many of the data do not have sufficient scientific support or explanation.

1) ! The authors do not prove the chemical form of silver in the obtained samples. This is unacceptable.

Moreover, I am surprised under the chemical formula “AgO”. It is repeated throughout the manuscript, so it is not a typo. Is it really silver(I,III)??? Then explain how the target was prepared and used, because this compound decomposes with the release of oxygen when heated (~100°C)!!

If silver oxide Ag2O was meant, then it is also unclear how it can be used in the method used (it also has low thermal stability).

In general, such chemical errors undermine the credibility of the authors.

2) The data provided (Supplemental Materials) shows that both Ag-containing materials are in the form of thin films rather than isolated nanoparticles (NPs). Thus, it is necessary to change the relevant terms throughout the manuscript or provide other evidence.

3) It is necessary to clearly determine the amount of silver introduced into all samples. For this, the authors can use MP-AES. Further, it will be informative to provide graph of the Ag release as a fraction of the total content.

4) !! On lines 511-515, the authors indicate "thicker CFx layer" for sample 5.3CFx/a-C:H:AgO than for sample 5.3CFx/a-C:H:Ag. This does not allow comparison of results methodologically!

It is necessary to redo the samples with the same thickness of the top layer.

Then, authors need to specify the thickness of each layer for all samples under study. Moreover, it is necessary to specify the accuracy with which the layer thickness can be controlled.

5) For clarity, a flowchart of the sample preparation sequence should be provided. It is not clear why the authors chose exactly 5.3% DC, and not 5.0 and not 5.5 or 5.7%, etc. For the rest of the samples, there is no such accuracy. Explain the idea, please.

6) There is no sufficient justification why the 5.3CFx/-samples were chosen for silver release and antibacterial research. What did the rest of the samples lose?

7) For clarity, the dynamics of silver release from samples with and without CFx layer should be presented on one graph. In addition, the current view of Figure 6a seems to require Ref. [3].

8) !!! The results of antibacterial studies do not correlate with the dynamics of silver release and do not have a clear explanation. In fact, it is not clear why 5.3CFx/ a-C:H:Ag, which gives a low silver content in solution, exhibits the highest antibacterial activity. At the same time, the 5.3CFx/ a-C:H:AgO sample, which gives a similar dynamics of silver precipitation, does not show any activity at all. It turns out that the dynamics of the release of silver does not play a role at all??

The available explanations are not satisfactory. Apparently, there is a methodological error, indicated in Comment 4. Thus, a number of samples should be removed from the study and new ones should be added instead.

9) In the Abstract, the authors emphasize the need to isolate silver within a few months. However, this issue is not considered in their own study. Accordingly, one of the aspects must be changed (correction of Abstract/Introduction or addition of long-term experimental data). The second aspect is preferable.

10) For clear comparison, antibacterial activity data should be provided for samples without silver, i.e. 5.3CFx/ a-C:H and a-C:H.

11) The authors should reduce the self-citation (currently 32%) and pay more attention to the literature on the silver release from composite materials with fluoropolymers. Now this aspect is covered poorly. Thus, the novelty of this work is not so clear.

12) The idea of O-F connections should be presented in more detail, supported by literature data. This also applies to the "oxygen-rich surface" effect on the deposition of CFx films.

13) It is necessary to expand the explanation for the appearance of silver at longer times of the ToF-SIMS depth profile (Fig. 5)

Comments on the Quality of English Language

Minor editing of English language required

Reviewer 2 Report

Comments and Suggestions for Authors

The authors have demonstrated the use of a porous CFx film containing Ag and AgO NPs as an additional top layer to elongate the release of Ag+ and increase the antibacterial activity.

The structural and optical characterizations demonstrate the successful preparation of CFx/a-C:H:Ag. Moreover, CFx/a-C:H:Ag has delayed the formation of biofilm and promoted the inactivation of bacteria by the disruption of their cell wall. Overall, this work can inspire more material design ideas of Ag NPs-based nanomaterials for antibacterial application. Therefore, I would like to recommend this work to publish in Nanomaterials. Below are some comments for the authors.

1. For a-C:H, a-C:H:Ag, and a-C:H:AgO, the photos of contact angles should be provided in the supporting information.

2. This paper would be more impressive, if the authors could provide the scanning electron microscopy (SEM) images of a-C:H, a-C:H:Ag, and a-C:H:AgO. The SEM images could provide the structural details of a-C:H, a-C:H:Ag, and a-C:H:AgO.

3. In Figure 7b, the descriptions of red arrow and circle should be provide in the caption.

4. For the introduction “Indeed, Ag NPs are known to be biologically active when they produce Ag ions (Ag+), exhibiting antibacterial activity against Gram-positive and Gram-negative bacteria”, more references could be cited to broaden the introduction.

https://doi.org/10.2147/IJN.S328767